# mHealth interventions for postpartum family planning in LMICs: A realist review

**Abinaya Chandrasekar**[1]*, **Emily Warren**[2], **Caroline Free**[1], **Judie Mbogua**[1], **Esther Curtin**[1], **Ursula Gazeley**[1], **Geoffrey Wong**[3], **Kathryn Church**[1], **Ona McCarthy**[1]

**1** Department of Population Health, The London School of Hygiene & Tropical Medicine, London, United Kingdom, **2** Department of Public Health, Environments and Society, The London School of Hygiene & Tropical Medicine, London, United Kingdom, **3** Nuffield Department of Primary Care Health Sciences, The University of Oxford, Oxford, United Kingdom

* abinaya.chandrasekar@lshtm.ac.uk

## Abstract

The unmet need for family planning is a pervasive public health concern in many low- and middle-income countries (LMICs). Mobile health (mHealth) interventions have been designed and implemented in LMIC settings to address this issue through health information dissemination via voice calls, apps, and short message services (SMS). Although the impact of mHealth programmes on postpartum family planning outcomes have been systematically reviewed, the contexts, conditions, and mechanisms underpinning programme engagement and their impact on outcomes remain unclear. This study aims to formulate hypotheses in the form of context-mechanism-outcome configurations (CMOCs) of whether, how, why, for whom, and in what contexts mHealth interventions implemented in LMICs influence postpartum family planning (PPFP) outcomes. We conducted a realist review of peer-reviewed and grey literature. Peer-reviewed literature was identified through MEDLINE, Embase, Global Health, Web of Science, and Google Scholar. Grey Literature was identified through The National Grey Literature Conference, FHI 360, Guttmacher Institute, Population Council, and MSI Reproductive Choices. Inclusion criteria were updated as the review progressed. Narrative data were analysed using dimensional analysis to build CMOCs. Two overarching concepts (underpinned by 12 CMOCs) emerged from the 37 included records: mobile phone access, use, and ownership as well as women's motivation. Women's confidence to independently own, access, and operate a mobile phone was a central mechanism leading to mHealth programme engagement and subsequent change in PPFP knowledge, awareness, and outcomes. Receiving family and social support positively interacted with this while low digital literacy and harmful gender norms pertaining to prescribed domestic duties and women's household influence were barriers to programme engagement. Intrinsic motivation for health improvement functioned at times both as a context and potential mechanism influencing mHealth programme engagement and PPFP outcomes. However, these contexts rarely occur in isolation and need to be evaluated as co-occurring phenomena. (Review registration: PROSPERO CRD42023386841).

**Data Availability Statement:** All relevant data are within the paper and its Supporting information Files.

**Funding:** The authors received no specific funding for this work.

**Competing interests:** The authors have declared that no competing interests exist.

## Introduction

Unintended pregnancies within short birth intervals–less than 24 months between delivery and a subsequent conception–are associated with higher risk of reproductive, maternal, neonatal, and child health (RMNCH) complications such as maternal anaemia, low birthweight, and pre-term birth [1, 2]. Sub-Saharan Africa and South Asia have the highest estimated child mortality burden (2.8 million and 1.5 million deaths of children under age 5 in 2019 respectively) [3]; postpartum family planning (PPFP) interventions addressing these contexts are more likely to result in improvements in infant and child mortality rates when they address the adverse consequences of short birth intervals [4]. The unmet need for family planning is the percentage of all married women who want to delay or limit future pregnancies but are not using any form of contraception [5]. In 2019, 218 million women in low- and middle-income countries (LMICs) had an unmet need for family planning (FP) methods and 49% of all pregnancies in LMICs were unintended [6].

Unmet need for family planning during the extended postpartum period is high in South Asia; 29% of South Asian women who had given birth in the previous year were not using LAM and had an unmet need for modern contraception [7]. A 2019 systematic review and meta-analysis reported high unmet need for modern contraception during the postpartum period in West Africa (59.4%, 95%CI: 53.4–65.4%) [8], which was attributed to misconceptions regarding the safety and lack of awareness of contraceptives; both issues can be addressed through timely availability of postpartum counselling on highly effective contraception [8]. Lack of access to effective contraceptive counselling during the postpartum period is common in LMICs and can lead to low knowledge and awareness regarding the benefits of postpartum family planning (PPFP) uptake [9]. Common barriers for PPFP uptake include misconceptions about modern contraceptives and a lack of knowledge regarding the limits of the lactational amenorrhea method [10]. Although the unmet need for family planning is also driven by sociocultural barriers such as gender norms, limited supply of contraceptives, and poor integration of family planning counselling into existing health services, the distal factor of PPFP knowledge is still a significant predictor of contraceptives uptake. This review focuses on programmes that aim to elicit PPFP behaviour change through improvements in knowledge and awareness of PPFP methods.

Globally, the widespread accessibility and affordability of mobile phones can be leveraged to overcome the social and behavioural barriers of PPFP uptake [11–13]. Mobile health (mHealth) interventions have grown in prevalence and their effectiveness has been evaluated using randomized controlled trials (RCTs) and systematic reviews. The mobile for reproductive health (M4RH) programme piloted in Kenya and Tanzania successfully improved participants' knowledge of FP services [14]. A 2019 RCT conducted in Kenya found that the two-way short message service (SMS) intervention group were 1.19 times (RR 95% CI 1.01–1.41, p = 0.04) more likely to use highly effective contraception compared to the control group [15]. The 'Kilkari' mHealth programme and RCT in Madhya Pradesh, India found that 38% of women in the intervention group reported use of modern reversible contraceptives compared to 31% of women in the control group; this finding was statistically significant [16, 17]. A systematic review of mHealth interventions in LMICs published in 2020 found that three out of the eight interventions were effective in improving contraceptive uptake and attributed these improvements to interactive and culturally tailored content including male partner involvement, motivational messages, and 'push' messaging from the intervention side to initiate engagement [18].

Several systematic reviews also report mixed evidence of the effectiveness of mHealth interventions in the RMNCH landscape on uptake and adherence of contraception and the

prevention of unintended pregnancies [19–23]. Smith et al. and Zulu et al. report a positive association between text-messaging interventions and oral contraceptive uptake across mostly high-income country (HIC) settings [19, 20]. Mangone et al.'s systematic review and content analysis of mobile applications concluded that several interventions were ineffective at reducing unintended pregnancies due to the promotion of contraceptive methods with low effectiveness [22]. However, Bassi et al. suggests that mHealth initiatives in India need to be evaluated with a focus on implementation factors specific to RMNCH outcomes [23]. Current evidence of the effectiveness of mHealth programmes mostly focuses on contextual variability across broad categories of age and geography. Four out of five trials included in Smith et al.'s quantitative synthesis were conducted in HICs [19]. Both Smith et al.'s and Zulu et al.'s Cochrane reviews included only RCT data [19, 20]. Few published studies explore the mechanisms underpinning the effective implementation of mHealth programmes and the contexts in which these mechanisms are activated [24–26]. mHealth programmes are complex social interventions and their success is often concurrently influenced by participant demographics, socially determined values and priorities, cellular network infrastructure, and interactions with healthcare services [27].

Realist Reviews were proposed as a methodological alternative to systematic literature reviews, that focus on a generative model of causality [28]. The overarching purpose of a realist review–or realist synthesis–is to review relevant available evidence on a topic and produce hypotheses of how contextual features and intervention resources interact and potentially activate mechanisms leading to various outcomes. Realist programme theories are theories of change and action, and change is explained through context, mechanism, and outcome configurations (CMOCs). CMOCs describe how specific contexts interact with individuals' reasoning (mechanism) to produce an outcome. Instead of pooling results of similar interventions to show an overall effect estimate, realist reviews aim to produce configurations of contexts, mechanisms, and outcomes of how intervention resources may 'work' under different conditions. Realist reviews often employ an iterative study design and include evidence from quantitative, qualitative, and mixed-methods studies as well as any relevant grey literature such as programme documents and reports [28]. Using a pre-determined inclusion criteria and a systematic search strategy is beneficial to initially narrow the scope of a realist review [29, 30]. The progressive focusing stage occurs as CMOCs emerge often in response to identified data 'gaps' and additional evidence specific to the identified context and mechanism interactions may need to be sought [31]. Although systematic reviews have previously reported whether mHealth interventions are effective in influencing RMNCH outcomes, a realist review offers the opportunity to examine how, why, and in what contexts these interventions lead to changes in PPFP outcomes. mHealth interventions use diverse delivery modes such as text-messages, voice-calls, or mobile applications. Evidence from these diverse mHealth interventions can be pooled using traditional systematic review methods if they report the same outcomes. Realist reviews, however, offer an understanding of why interventions with similar outcomes can produce inconsistent results and explain their differential impacts. Evidence on mHealth interventions are often varied and not solely peer-reviewed publications; realist syntheses are better suited to reviewing documents with this level of heterogeneity because the iterative study design provides flexibility in selecting evidence for synthesis regardless of methodological rigour. In realist reviews, studies are included on the basis of relevance to theory building; transparency of the rigour of included documents is still a key reporting criterion [32] to contextualise findings from the reviewed evidence.

This realist review occurred in conjunction with and was informed by a realist evaluation of an automated voice-messaging programme based in India ('mMitra') which aims to improve socioeconomically vulnerable women's access to pregnancy and postpartum care information.

Further details on the design and implementation of the mMitra programme is provided in S1 Text.

### Aims and objectives

**Aim.** To formulate CMOCs explaining whether, how, why, for whom, and in what contexts mHealth interventions implemented in LMICs influence PPFP outcomes.

**Objectives.**

1. Develop an initial programme theory of whether mHealth interventions targeting PPFP outcomes in LMICs work, how, why, for whom, and in what contexts through exploratory searches of a sample of peer-reviewed and grey literature and expert consultation.

2. Conduct a realist review of peer-reviewed and grey literature on mHealth interventions implemented in LMICs using a systematic search strategy.

3. Refine the initial programme theory using evidence identified in the realist review to produce a series CMOCs explaining whether mHealth interventions targeting PPFP outcomes in LMICs work, how, why, for whom, and in what contexts.

## Materials and methods

The review protocol was registered with PROSPERO (CRD42023386841) and findings from this review were reported in line with realist synthesis publication standards [33].

### Initial Programme Theory (IPT)

We conducted an exploratory scoping of peer-reviewed and grey literature that detail the development or implementation of mHealth interventions targeting PPFP outcomes in LMICs. The first relevant document to be identified was the Mobile Alliance for Maternal Action (MAMA) report detailing the implementation of the MAMA approach in four LMIC settings [34]. The MAMA series of interventions, of which mMitra is a part, was an appropriate starting point for scoping literature to develop the IPT because of the scope of this project, in which the realist review detailed in this manuscript forms the basis of a realist evaluation of the mMitra programme. The review and evaluation components were designed to inform each other. The mMitra programme was included in the MAMA report and informed the additional scoping of affiliated mHealth programmes. The 'Kilkari' programme was identified as a relevant contributor to the initial programme theory building process. Lefevre et al.'s 2022 publication of a 'Kilkari' evaluation using an RCT study design was hand-searched and included in the review. An additional report of the 'Aponjon' programme published for the mHealth Compendium special edition of 2016 was included in this stage of the review. We conducted in-depth readings of the three documents and used dimensional analysis to analyse narrative data from the included documents and to extract context, mechanism, and outcome configurations (CMOCs) that formed the initial programme theory [35]. Dimensional analysis is a variant of grounded theory involving the inductive and structural coding of contexts, conditions, processes, and consequences of data to build underlying theories [35]. CMOCs were diagrammatically constructed using Dalkin et al.'s framework [36] (S1 Fig) to better aid discussions of how intervention resources can potentially transform participant reasoning when interacting with specific contexts. The research team was consulted during the IPT development process to check emerging theories.

## Search and screening

To initially narrow the scope of the review, we used a systematic search strategy and pre-determined inclusion criteria. We produced a systematic search strategy (S1 Table) in consultation with an information specialist at The London School of Hygiene & Tropical Medicine. The systematic search of peer-reviewed and grey literature was conducted in December 2022. Peer-reviewed literature was searched on MEDLINE, Embase, Global Health, Web of Science, and Google Scholar. Grey literature was identified from The National Grey Literature Conference, Population Council, Guttmacher Institute, MSI Reproductive Choices, FHI 360 as well as a targeted 'PDF' file only searches using Google's search engine. Two additional iterative searches of peer-reviewed literature were conducted to identify more qualitative studies and mechanism-specific evidence for theory refinement respectively (S1 Table). All titles and abstracts were double screened on Rayyan [37] and a random 10% sample of full texts were double screened [31, 33]. All conflicts were resolved through discussion. All double screening was blinded on Rayyan. Double-screening a random 10% sample of full texts is common practice in realist reviews [38, 39]. Realist reviews are a form of configurational review. If any documents with relevant data were to be missed, it is unlikely to introduce significant threats to the validity of the findings. When combined with double-screening 100% of titles and abstracts, these processes ensure that the inclusion criteria were verified and applied suitably. Hand-searched documents for the progressive focusing stage of the review were single-screened. Table 1 outlines the eligibility criteria for the systematic search phase of the review. Interventions exclusively developed for and delivered by mobile phones were included. Only interventions targeting postpartum family planning outcomes in women within one year postpartum were included. All study designs and methodologies were eligible for inclusion but studies with insufficient detail required for CMOC construction and theory building were excluded from the review.

We carried out two additional iterations of literature searches. Firstly, a systematic search of peer-reviewed and grey literature on MEDLINE was conducted to identify only qualitative studies. Following consultation with co-authors and given the iterative nature of realist reviews, qualitative studies were hypothesised to be more relevant to the construction of

**Table 1. Eligibility criteria for systematic search phase of realist review.**

| Inclusion criteria | Exclusion criteria |
|---|---|
| Interventions delivered in LMICs | Interventions developed and delivered through internet websites, although accessible through smartphone web browsers. |
| All study types and methodologies (qualitative, quantitative, and mixed methods) | Interventions targeting women pre-conception or after > 1 year postpartum. |
| All intervention and programme enrolment settings (e.g. hospitals, community recruitment, or remote interventions) | Interventions exclusively targeting sexual and RMNCH outcomes that are not relevant to PPFP. (e.g., childhood vaccination schedules or neonatal complications) |
| Outcomes targeted towards/measured in pregnant women and women who are ≤ 1 year postpartum. Study participants can be anyone (e.g., women receiving the programme, healthcare workers, field workers etc.) | Insufficient data/information to construct CMOCs. |
| Interventions delivered exclusively through mobile phones. | |
| Interventions that address PPFP outcomes (e.g. knowledge of PPFP methods, PPFP uptake, adherence to PPFP methods, and prevalence of unintended pregnancies). | |

CMOCs due to the theoretical nature of this study. During the progressive focusing stage of the review, specific documents were manually searched on Google Scholar using a combination of search terms including 'phone', 'share', 'mHealth', 'husband', 'stage-based intervention', 'knowledge', and 'motivation'. These search terms were identified from Cochrane systematic reviews in this topic area and framed using Boolean operatives to ensure usefulness on Google Scholar [19, 40–42]. Both systematic searches of peer-reviewed literature were screened using the eligibility criteria in Table 1. The eligibility criteria were not applied to manual-searching during the progressive focusing phase as the purpose was to solely identify empirical evidence of contexts in which the mechanisms of interest are activated, which does not need to be limited by the specific outcome of interest (PPFP).

## Data extraction and synthesis

All included documents were exported from Rayyan, and narrative data were extracted and analysed using NVivo V1.7.1 [43]. CMOCs were developed using a realist logic of analysis which involved a combination of the dimensional analysis and Dalkin et al.'s framework for CMO refinement. Preliminary lists of contexts, conditions, processes, and consequences from each programme in the review were developed through multiple readings of included documents and line-by-line coding [44]. All coded contexts, conditions, processes, and consequences were categorised and recoded into contexts, mechanisms, and outcomes to align with the CMOC heuristic employed in realist studies. Additional iterative searches of literature helped refine and code data into CMOCs. Final programme theories were proposed by the primary researcher then reviewed and sense-checked by three additional researchers.

All included studies were assessed for relevance and rigour, in line with realist review implementation guidelines [31]. We used the mixed-methods appraisal tool [45] to assess the methodological rigour of all included texts that reported data from the following study types: qualitative studies, RCTs, observational studies, and mixed-methods studies. We did not differentially weight studies for inclusion in this review on the basis of methodological rigour. To assess the methodological quality of texts describing mHealth interventions without a formal study design, we used the mobile health evidence reporting and assessment (mERA) checklist produced by the WHO mHealth Technical Evidence Review Group [46]. The mERA checklist also complemented the realist approach of this study as it included identification of barriers and facilitators of programme implementation, which are relevant for identifying CMOCs to build realist programme theories. The quality appraisal scores of all included texts will be discussed in detail in the 'Quality Appraisal' section.

## Results

### Search and screening

A total of 1,086 titles and abstracts were screened, after deduplication in Endnote [47]. After the title and abstract screening stage, 1,024 texts were excluded. AC checked references of all included systematic reviews to identify additional relevant texts. Following this snowballing process, 84 full texts were screened by AC and OM. In both title and abstract and full text screening stages, we discussed conflicts until a unanimous decision was achieved. There were two discrepant screening decisions identified during double-screening of the random 10% sample of full-texts. Hand searching during the progressive focussing stage of the review identified an additional 20 texts. 37 texts were included in the final review. The PRISMA flowchart outlining this process can be found in Fig 1.

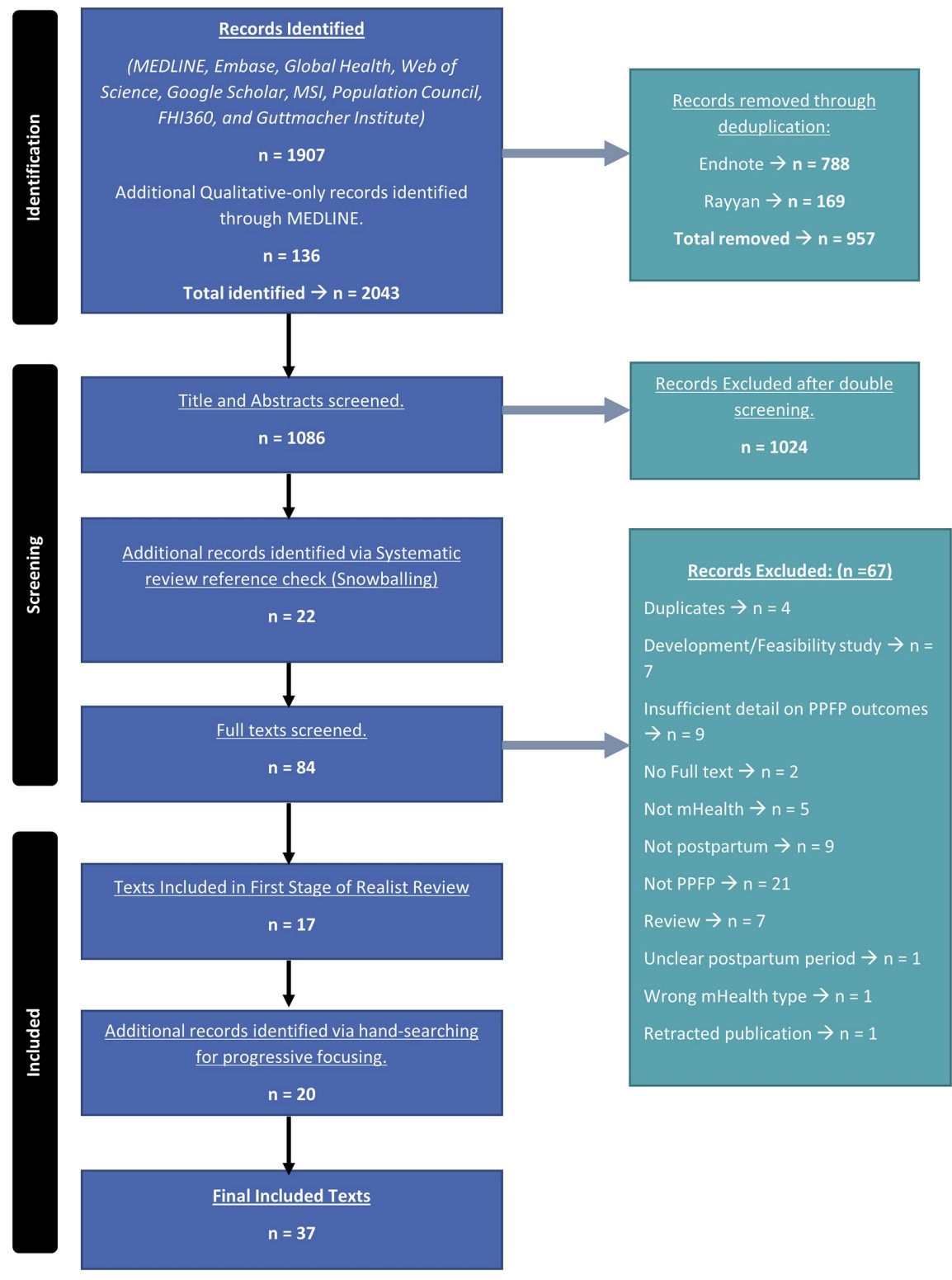

**Fig 1. PRISMA flow diagram of screening process.**

### Included records

Included records represent mHealth evidence from several LMICs. India (n = 11) and Kenya (n = 7) were the most frequent LMIC settings for mHealth implementation in the review sample. A combination of qualitative (n = 9), quantitative (n = 24) and programme documents (n = 4) were included in the review. RCTs were the most frequent study design (n = 9) followed by qualitative studies using in-depth interviews (n = 9). Characteristics of all included documents are summarized in Table 2 below.

### Quality appraisal

Detailed relevance and rigour scores are presented in S1 Data. Most included documents scored highly on MMAT and mERA assessments of rigour. Relevance scores were high for most included documents. Low methodological rigour was attributed to some included grey literature, such as programme documents, because of lack of transparency in the reporting of fidelity measures, cost assessments, and the replicability of reported findings [51, 54]. In the academic literature, studies that included inadequate randomisation, poor adherence to the assigned intervention, inappropriate qualitative methods to address the research questions, and completeness of the outcome data were considered as having low methodological rigour [53, 60, 66, 67]. Manually searched documents identified during the progressive focusing stage were more likely to have low relevance scores because they only provided insight on contexts and mechanisms, but not outcomes. One study was removed from consideration in this review because of a journal retraction notice [81]. The implications of the rigour assessment of included evidence is further detailed in the 'Strengths and Limitations' section of this review.

### Main findings

#### Phone access, use, and ownership

Mechanisms in Tables 3 and 4 are disaggregated into resources and reasoning, in line with Dalkin et al.'s framework [36], to better represent the impact of intervention components or participant resources on outcomes when interacting with specific contexts. Each realist causal explanation is presented in CMOC format in the fourth column of Table 3.

**Barriers of mHealth programme engagement.** Phone access, use, and ownership was a central theme of mechanisms influencing mHealth programme engagement and subsequent PPFP outcomes among LMIC women. The primary outcome explored in this realist review is engagement with mHealth programmes, which subsequently influences the more distal outcomes of PPFP knowledge, awareness, or uptake. This is because mHealth programmes, per se, are unlikely to increase PPFP knowledge, awareness, or uptake if women do not engage with them. Hence engagement by women is an important first step when trying to understand the use of mHealth interventions for this population. It is also important to acknowledge that knowledge or awareness are key domains in health behaviour change models, and can potentially impact PPFP behaviour change, although not the sole determinant of outcomes associated with PPFP uptake [83, 84]. The context of women with low digital literacy often interacted negatively with mHealth programme content because of their unfamiliarity with operating mobile phones and led to limited engagement with the mHealth programme [CMOC1] [34, 49, 52, 55, 58, 59, 74, 75, 82]. Within Khatun's study population of households in the Chakaria district of Bangladesh, approximately 50% of individuals did not know how to send and receive text messages [82]. Although Bihari women enrolled in Kilkari had high independent ownership of mobile phones, these women rarely had SIM cards 'registered in their names' or independently recharged their phones with 'credit' [52].

**Table 2. Characteristics of all records included in the review.**

| Author and Study Location | Year | Study Design | Intervention Description | Identified concepts that contributed to CMOC development |
|---|---|---|---|---|
| | | | *Initial Systematic Search of Peer-reviewed and Grey Literature* | |
| Carmichael et al. [48] India | 2019 | Cluster randomized controlled trial | ICT-CCS Tool provided mobile phone-based job aids for FLWs to improve the coverage, quality and coordination of their service to pregnant and postpartum women, Frontline workers and women within 1 year postpartum in Saharsa district (Bihar, India). | Dispelling myths and rumours on PPFP methods |
| Dev et al. [49] Kenya | 2019 | Qualitative study–semi-structured interviews | Client facing mobile application providing systematic and personalized contraceptive counselling to postpartum women and guiding women through decision-making regarding FP, Postpartum women, and FP providers (nurses) from 4 Kenyan maternal and child health clinics across 2 rural sites(Kisumu and Siaya Counties) and 2 urban sites in Nairobi. | Adolescent and young women, decision aid mobile app, dispelling rumours on contraception, perceived safety of methods during breastfeeding, dissemination of accurate and trustworthy information. |
| Dyer [50] Timor-Leste | 2015 | Qualitative study–in-depth interviews | Women with a minimum of 2 children who had recently completed participation in the 'Liga Inan' intervention, Mobile health (mHealth) technology—health information dissemination and a line of communication with midwives to empower participants to make informed decisions about their and their babies' health and facilitate improved links with health facility staff resulting in an uptake of services. | History of military occupation and forced sterilisation, distrust of healthcare institutions, discussing family planning with spouses, access to knowledge, improved motivation to use healthcare services, birth spacing knowledge. |
| mHealth Compendium special edition [51] Bangladesh | 2016 | Programme document/report | Women between 6 and 42 weeks of pregnancy and mothers with a child under one year of age, voice/text, and call centre counselling. | Gatekeepers, mobile network operators, local dialect, phone sharing with spouses, requiring and receiving husband's approval for phone use, pregnancy not viewed as a health issue, PPFP knowledge and awareness, birth spacing outcomes. |
| GSMA [52] India | 2016 | Programme document/report | Pregnant women and women within 1 year postpartum in India, Kilkari (a baby's gurgle in Hindi) delivers free, weekly, time-appropriate audio messages about pregnancy, childbirth, and childcare directly to families' mobile phones from the second trimester of pregnancy until the child is one year old. Kilkari seeks to increase the capacity of pregnant women, new mothers, and their families to adopt healthier behaviours, through increasing their knowledge, shifting attitudes, and building self-efficacy. The objective is to improve family health–including family planning, reproductive, maternal, neonatal and child health, nutrition, sanitation, and hygiene—by generating demand for healthy practices. | Free cost, women's phone ownership, audio only messages, targeting gatekeepers, requiring and receiving husband's approval for phone usage, pregnancy viewed as a natural experience and not requiring intervention, PPFP knowledge, awareness, outcomes, birth spacing outcomes. |
| Gupta et al. [53] India | 2018 | Pre-post study–acceptance of PPFP methods following enrollment in mHealth reminder/counselling programme | Patients attending pregnancy and postpartum checkups at S.N. Medical College and Hospital in Agra, Uttar Pradesh, India, Phone calls and reminder calls made by counsellor to encourage PPFP adoption among pregnant and postpartum women attending ANC/PNC checkups at the study hospital. | Telephone counselling, rural environment, interaction with healthcare providers, phone call reminders, |

(*Continued*)

**Table 2.** (*Continued*)

| Author and Study Location | Year | Study Design | Intervention Description | Identified concepts that contributed to CMOC development |
|---|---|---|---|---|
| Harrington et al. [15] Kenya | 2019 | Randomized controlled trial | Pregnant women attending public hospitals in Kisumu and Siaya counties in Western Kenya, automated SMS and SMS dialogue between participants and clinicians—Participants indicate their language of choice (English, Kiswahili, or Dholuo), preferred name, and preferred day and time to receive automated messages (Theory of planned behaviour). | 2-way SMS messaging, couple level intervention, engaging gatekeepers, sharing messages between spouses, sharing messages at the community level, postpartum decision-making related to PPFP |
| Jacaranda Health [54] Kenya | | Programme document/report | Pregnant women and women within 1 year postpartum attending the Jacaranda Health facility in Kenya, gestation-specific tips and reminders for appointments, helpdesk service answering questions, mother receives newborn care tips and immunization appointment reminders. | Information provision, empowering women with agency to improve self-care, uptake in postpartum family planning |
| Jones et al. [55] Kenya | 2020 | Randomized controlled trial | Postpartum women who delivered at study sites in Kenya and had access to a cell phone in which they could receive SMS messages, SMS messages in Swahili—free for participants, two-way messages, messages delivered using international guidelines and guided by clinicians at Jacaranda Health, postpartum checklist, general postnatal care, and family planning. | SMS nudge messaging, 2-way communication, high baseline awareness of postpartum care, no motivation to change behaviours, PPFP uptake, choice of PPFP method (modern, traditional, long-acting reversible etc.), failure to attend rates for postpartum checkups. |
| Lefevre et al. [17] India | 2022 | Randomized controlled trial | Women > = 18 years old, 12–34 weeks gestation, could speak and understand Hindi, and owned or had access to a mobile phone during the day when Kilkari calls were likely to come, Kilkari—90 minutes of content delivered via 71 once weekly voice calls (24 during pregnancy, 24 within the first 6 months postpartum and 24 from 7 to 12 months postpartum. Individual calls span an average of 77 s in duration and are framed as coming from 'Dr Anita'). 18% of total programme content is on family planning, benefits of FP, modern reversible methods, sterilization, and pregnancy tests. | Indian women's limited access to mobile phones, phone sharing practices, having sons and their impact on decision making surrounding family planning, reversible methods versus permanent methods, higher programme effectiveness in vulnerable communities, higher PPFP usage in women with sons. |
| Maslowsky et al. [56] Ecuador | 2016 | Prospective evaluation–randomised allocation of intervention and control groups. | Mothers who are > = 15 years of age, speak Spanish and their newborn had not been admitted to the neonatal intensive care unit, educational session administered by the nurse via phone within 48 hours of hospital discharge and access to a nurse on call during the first 30 days of the newborn's life. | High mobile phone penetration in Ecuador, higher motivation and interest to pursue family planning, limited access to contraception despite an interest in using it, higher use of HEC methods in intervention group compared to controls, overall contraceptives usage not significantly different between intervention and control groups. |
| McConnell et al. [57] Kenya | 2018 | Randomized controlled trial | Pregnant women attending antenatal care at Jacaranda Health in Kenya (provides maternal and newborn healthcare to poor urban women), Voucher to use LARC and SMS reminder to redeem vouchers sent to eligible participating women attending Jacaranda Health maternity clinic. | Voucher for contraceptives, appointment reminders, desire for privacy regarding FP decision making, PPFP uptake |

(*Continued*)

**Table 2.** (Continued)

| Author and Study Location | Year | Study Design | Intervention Description | Identified concepts that contributed to CMOC development |
|---|---|---|---|---|
| Murthy et al. [58] India | 2020 | Quasi-experimental study–non-randomized trial | Pregnant women who spoke Marathi or Hindi from F North and M East wards in Mumbai, mMitra—145 audio messages designed with BabyCenter discussing pregnancy and postpartum care topics, messages delivered twice a week roughly, call centre feature to reconnect with mMitra if dropped or for enquiries. | Automated voice messaging intervention, low literacy, low digital literacy, employment status of women, domestic labour demands, familial support, tailored messages building trust and rapport with healthcare services, PPFP knowledge, awareness, and outcomes + birth spacing. |
| Scott et al. [59] India | 2021 | Qualitative study–in-depth interviews and family/group discussions | Households enrolled in Kilkari RCT with very high to medium listenership of the Kilkari programme, Kilkari (a baby's gurgle in Hindi) delivers free, weekly, time-appropriate audio messages about pregnancy, childbirth, and childcare directly to families' mobile phones from the second trimester of pregnancy until the child is one year old. Kilkari seeks to increase the capacity of pregnant women, new mothers, and their families to adopt healthier behaviours, through increasing their knowledge, shifting attitudes, and building self-efficacy. The objective is to improve family health–including family planning, reproductive, maternal, neonatal and child health, nutrition, sanitation, and hygiene—by generating demand for healthy practices. | Call content aligns with existing practices, social norms, and personal worldviews, call content repetition, engaging with common misconceptions, provide clarification of common misconceptions of FP, targeting men and women as joint FP decision makers, satisfaction with traditional FP methods, birth spacing desires, desires of limiting future pregnancies. |
| Scott et al. [16] India | 2021 | Qualitative study–in-depth interviews | Women enrolled in Kilkari who showed very high, high, and medium listenership based on the percentage of cumulative Kilkari call content that was picked up and allowed to play before being hung up, Kilkari—90 minutes of content delivered via 71 once weekly voice calls (24 during pregnancy, 24 within the first 6 months postpartum and 24 from 7 to 12 months postpartum. Individual calls span an average of 77 s in duration and are framed as coming from 'Dr Anita'). 18% of total programme content is on family planning, benefits of FP, modern reversible methods, sterilization, and pregnancy tests. | Dissatisfaction with traditional FP methods, husband listened to call content, phone sharing, perceptions of harm from modern HEC methods, husband takes phone away during work hours, listening to calls with husband, did not pay close attention to calls, joint decision-making regarding FP, busy with work or without access to phone and did not listen to FP calls. |
| Unger et al. [60] Kenya | 2018 | Randomized controlled trial | Pregnant women, at least 14 years old with access to a mobile phone and able to read SMS were eligible for participation—women seeking ANC care at an MCH were enrolled, Mobile WACh XY intervention—one-way messages, or two-way messages, available in English or Kiswahili at times and days of the week preferred by the participant, topics include ANC, pregnancy complications, family planning, EBF, immunisation, and visit reminders, free of charge to participants. | Personalized messaging, two-way messages, high motivation for health-seeking in trial population, engagement with programme, uptake of early PPFP. |
| USAID et al. [34] Bangladesh, South Africa, India, Nigeria | 2016 | Programme document/report | MAMA intervention implementation—lessons learned from four programmes in Bangladesh, South Africa, India, and Nigeria, Stage-based delivery of pregnancy and postpartum messages–up to 1 year postpartum, aligning with gestational age = improved motivation and increases likelihood of behaviour change. | Stage-based messaging, SMS messages vs. voice calls, rural vs. urban settings, free of cost service vs. paid model, push vs. pull messaging, engagement with the programme, response to tailored messaging, improved motivation, uptake of self-care measures during pregnancy and postpartum period. |
| *Progressive Focusing* | | | | |
| Bongaarts et al. [61] N/A | 1990 | Opinion article | Family planning programmes in LMICs | Low motivation, family and social opposition to FP uptake, motivation for PPFP uptake. |

**Table 2.** (Continued)

| Author and Study Location | Year | Study Design | Intervention Description | Identified concepts that contributed to CMOC development |
|---|---|---|---|---|
| Cleland et al. [62] N/A | 2015 | Evidence review of Postpartum contraception data. | Family planning programmes in LMICs, antenatal and postnatal care interventions addressing FP. | Awareness of risks associated with non-use of PPFP, legality of medical abortion provision, motivation for PPFP uptake. |
| Doron [63] India | 2012 | Qualitative study–observations and interviews. | Families and women with mobile phones in Varanasi, India. | Family shared phones, family dynamics, women's agency within spouse's family, tension between family members, low mobile phone access for women. |
| Fjeldsoe et al. [64] N/A | 2009 | Systematic Review | Individuals enrolled in behaviour change interventions delivered by SMS and use pre-post assessment for evaluation, Studies were included in the review if they (1) evaluated an intervention delivered primarily via SMS, (2) assessed change in health behaviour using pre–post assessment, and (3) were published in English in a peer-reviewed scientific journal. | Tailored messages, SMS mHealth interventions, participant engagement, behaviour change (not specific but applicable to PPFP behaviour change) |
| Frenn et al. [65] N/A | 2003 | Pre-post intervention study | Low-income, culturally diverse students from an urban middle school, stage-based interventions for low-fat diet with middle school students. | Timing of interventions, tailoring interventions to stage of motivation/readiness/willingness for behaviour change |
| Harvey et al. [66] USA | 2002 | Qualitative study–semi-structured interviews. | The women and their male partners were recruited from family planning and STD clinics and other community locations in each city using both passive (e.g., posters and fliers) and active (e.g., recruiters approaching potential participants in the clinics) strategies. | Joint decision-making, spousal dynamics, higher reported use of FP. |
| Jareethum et al. [67] Thailand | 2008 | Pre-post intervention study. | Healthy pregnant women attending antenatal clinic at Siriraj Hospital, Thailand, the study group received two SMS messages per week from 28 weeks of gestation until giving birth. The other group was pregnant women who did not receive SMS. Both groups had the same antenatal and perinatal care. | Family support, stage of gestation, high confidence in pregnancy care, low anxiety associated with pregnancy. |
| Kazi et al. [68] Kenya | 2017 | Survey study | 8 health facilities in Northern Kenya as part of a program to scale up an mHealth service in rural and remote regions. The study was conducted at 6 government health facilities in Isiolo, Marsabit, and Samburu counties in remote and northern arid lands (NAL). Two less remote health facilities in Laikipia and Meru counties in more populated central highlands were included as comparison sites, text messaging (short messaging service, SMS)-based mHealth intervention for improvements in antenatal care attendance and routine immunization among children in Northern Kenya. | Shared phone access, social trust and support |
| Khatun [69] Bangladesh | 2016 | Descriptive study–quantitative survey. | A total of 4915 randomly selected household members aged 18 years and over completed the survey—survey conducted in the Chakaria sub-district of Bangladesh from November 2012 to April 2013. | Female phone ownership and operation, seeking permission to use mobile phones/seek healthcare through phones, knowledge of SMS, ability/confidence to seek healthcare information, trusting mHealth services. |

*(Continued)*

**Table 2.** (*Continued*)

| Author and Study Location | Year | Study Design | Intervention Description | Identified concepts that contributed to CMOC development |
|---|---|---|---|---|
| Kiene et al. [70] Uganda | 2014 | Quantitative study–cross-sectional surveys | The data was collected from pregnant women attending the antenatal clinic (ANC) at Gombe Hospital. This study was part of a larger study examining partner attendance at ANC and the uptake of partner HIV testing and use of contraceptives. Women typically attend ANC for the first time at the 4th month of their pregnancy and then approximately every month until the 8th month, at which point they return every 1 to 2 weeks. After delivery, the women return at 6 weeks for post-natal care and then at 10 and 15 weeks for infant immunisations, Participants (N = 301) completed a baseline questionnaire interview and a follow-up questionnaire interview approximately 10weeks postpartum, although this time varied based upon when mothers brought their infants to the hospital for immunizations. | Differences in future pregnancy planning, joint decision-making, perceptions of partner's attitudes related to FP. |
| Kocher et al. [71] American Samoa | 2018 | Qualitative study–semi-structured interviews. | A convenience sample of 18 women who had given birth to a child in the past 12 months (Samoan ethnicity, 18 years old at the time of recruitment) were recruited into the study during their visits to the Well Baby Clinic at the Tafuna Family Health Centre in Tafuna, American Samoa. | Perceiving pregnancy as an illness, increased contact with health systems, receiving family support during pregnancy, prioritizing childcare over self-care, empowered, feeling taken care of and supported to make self-care decisions, behaviour change—adoption of self-care behaviours during pregnancy and postpartum period. |
| Lawrence et al. [72] N/A | 2007 | Literature Review | Pregnant women with smoking behaviours (current or past), smoking cessation interventions for pregnant women attempting to change behaviours using the stage of change model—stage-based intervention related to periods of high motivation. | Behaviours that are resistant to change (pleasurable/automatic/addictive), intervention timing, intrinsic and extrinsic motivation, smoking cessation during pregnancy. |
| Lopez et al. [73] N/A | 2016 | Systematic review | Brief educational interventions for improving contraceptive use among young people that are feasible for implementing in a clinic or similar setting with limited resources—The intervention had to be sufficiently brief for a clinic, i.e., one to three sessions of 15 to 60 minutes plus potential follow-up. The strategy had to emphasize one or more effective methods of contraception. Primary outcomes were pregnancy and contraceptive use, RCTs | Follow-up incorporated into interventions, length of intervention/delivery, motivation, OC pills uptake/behaviour change. |
| Messinger et al. [74] Bangladesh | 2016 | Qualitative study–interviews | MR clients, formal MR providers, and informal MR providers in four low-income settlements in the Dhaka and Sylhet districts of Bangladesh, mHealth interventions—broadly speaking—exposure to mHealth by MR clients —knowledge, attitudes and practices regarding mHealth of both MR clients and formal and informal sexual and reproductive healthcare providers in urban and rural low-income settlements in Bangladesh. | Low access to MR services, information dissemination, incorporation of in-person services, motivation to seek MR services. |

(*Continued*)

**Table 2.** (Continued)

| Author and Study Location | Year | Study Design | Intervention Description | Identified concepts that contributed to CMOC development |
|---|---|---|---|---|
| Mohan et al. [75] India | 2020 | Quantitative study–survey analysis. | Data drawn from the 2015 National Family Health Survey (NFHS) in India included a national sample of 45,231 women with data on phone access. Survey design weighted estimates of household phone ownership and women's access among different population subgroups are presented. | Joint decision-making, mobile phone access, urban vs. rural residence and gender gap in mobile phone access, ability to access/use mobile phone. |
| Molla et al. [76] Ethiopia | 2007 | Quantitative survey study | Youth population (15–24 years) living in the study catchment area in rural Ethiopia—interviewed about previous condom use and future planned behaviours regarding contraception. | Previous experience, behavioural intention, intrinsic motivation, positive perceptions, efforts to maintain and carry out intentions. |
| O'Brien et al. [77] Ireland | 2017 | Qualitative study–cross-sectional interviews. | Pregnant women (n = 22), early pregnancy Body Mass Index > 25 kg/m2 | Pregnancy as a stimulant for behaviour change, childcare vs self-care, intrinsic and extrinsic motivation. |
| Park et al. [78] USA | 2008 | Randomized, treatment-control design with pre-post intervention assessments. | Convenience sample of 160 young adults (aged 18–24) recruited by community educators in 4 states. Study completers (n = 96) included a mix of racial/ethnic groups and family demographics but were predominantly white females without children, A Transtheoretical Model (TTM)-based, stage-tailored Internet program, F&V (Fruit & Vegetable) Express Bites, was delivered to treatment group participants; controls received nontailored messages in a comparable format. | Tailoring content to individuals' stage of motivation, perceived vulnerability to adverse health outcomes. |
| Schuler et al. [79] Tanzania | 2011 | Qualitative study–in-depth, open-ended interviews and focus groups. | Young currently married men, 30 young married women and 12 older people who influenced FP decisions. | Previous experience, gender and misinformation, fears of MCM side effects, male involvement, benefits of birth spacing outweighed negatives. |
| Steenson et al. [80] India | 2008 | Qualitative study–ethnography | Ethnographic study of mobile phone sharing in Bangalore, participants include women and households in Bangalore, Interviews to assess mobile phone access and usage. | Concealing shared mobile phone usage, gender norms, unequal sharing of mobile phones. |

Programmes that used scheduled automated voice-calls to disseminate information were ineffective at engaging women, specifically in women who were occupied with domestic duties in the household [16, 58, 59, 74, 82]. These women felt they must prioritise their commitment to domestic labour over answering phone calls during the day which led to poorer engagement with their respective mHealth programmes [CMOC 2] [16, 58,59, 74, 82]. Scott et al. reported that women used mobile devices scarcely due to being 'busy with work. . .like taking care of the children' or 'cleaning the house' [59]. However, some studies still reported the benefits of using voice-messaging to deliver health information over text messages because they were potentially more usable by low-literacy populations. Murthy et al. suggest that voice-messages are a more accessible format for low-literacy populations compared to text-messages as a means of potentially improving PPFP knowledge and use [58]. Bangladesh's MAMA-informed programme, Aponjon, found that rural populations and women of low socio-economic status had an unmet need for voice calls to receive health information and potentially yield improved PPFP use due to functional and digital literacy concerns [34].

Low engagement with mHealth programmes was also a result of phone sharing practices between women and their husbands, mostly informed by traditional gender norms. In

**Table 3. A summary of the CMOCs describing the influence of phone access, use, and ownership on mHealth intervention implementation in LMICs targeting PPFP outcomes.**

| CONTEXT (C) | MECHANISM (M) | | OUTCOME (O) | CMOC |
|---|---|---|---|---|
| | **Resources** | **Reasoning** | | |
| **CMOC 1**<br>Women with low digital literacy | mHealth programme content | Unfamiliarity with operating mobile phones *(Barrier)* | Little to no engagement with mHealth programme | For women with low digital literacy (C), mHealth programme content was not effective in motivating behaviour change (O) because they were unfamiliar with operating mobile phones and could not effectively engage with the mHealth programme (M). [34, 49, 52, 55, 58, 59, 74, 75, 82] |
| **CMOC 2**<br>Women are occupied with domestic duties for a large part of the day. | mHealth programmes using automated-voice calls scheduled during the day | Women feel they must prioritise domestic duties over answering phone calls. *(Barrier)* | Little to no engagement with the mHealth programme | Women who are occupied with domestic duties during the day are expected to engage with mHealth programmes through use of voice calls (C) but because they feel they must prioritise domestic duties over answering phone calls (M) it leads to little or no engagement with the mHealth programme (O). [16, 58, 59, 74, 82] |
| **CMOC 3**<br>Woman receives family and social support to operate and own a mobile phone independently. | mHealth programme content directed to woman (Woman is the intended audience of the messages) | Women feel confident to independently own and operate a mobile phone and participate in the mHealth programme *(Facilitator)* | High engagement with the mHealth programme | When an mHealth programme's content is directed exclusively to women and women receive family and social support to operate and own a mobile phone independently (C), women feel more confident to participate in mHealth programmes (M) and can lead to high engagement with the programmes (O). [52, 53, 55, 59, 74, 75, 80, 82] |
| **CMOC 4**<br>Society values women who exert minimal influence over household decision making | Women share mobile phones with their spouses, relatives, or household. | Women cannot negotiate or influence physical access of a shared mobile phone *(Barrier)* | Little to no engagement with the mHealth programme | Little to no engagement with mHealth programmes (O) may occur in societies where women are subservient and do not have independent access to mobile phones (C) because they cannot negotiate or influence physical access to a mobile phone (M). [16, 17, 34, 52, 53, 59, 63, 68, 80, 82] |
| **CMOC 5**<br>Society values male dominance over female partners | mHealth programme content directed at women | Men fear consequences of independent mobile phone usage by wives and gatekeep access to a shared mobile phone *(Barrier)* | Little to no engagement with the mHealth programme | When mHealth programme content is directed at women in societies that value female subservience and male dominance in partnerships (C), men fear the negative consequences of women's independent mobile phone usage and gatekeep their access to shared mobile phones (M) which leads to little or no engagement with an mHealth programme. [15, 52, 53, 57, 59, 63] |

societies that value women's subservient role in household decision-making, women who share phones with their family members are unable to negotiate physical access of shared mobile devices which leads to little or no engagement with their specific mHealth programme [CMOC 4] [16, 17, 34, 52, 53, 59, 63, 68, 80, 82]. Although infrequent within the study population, Scott et al. reported observations of women being 'barred from handling the husband's phone' [16]. Women often provided their husband's numbers–which are not accessible during work hours while husbands are away from the household–during mHealth programme enrolment which negatively impacted their engagement with the programme content [53]. A 2011 impact evaluation of the Aponjon programme found that most 'women shared their phones

**Table 4.** A summary of CMOCs describing the influence of mHealth intervention implementation in LMICs targeting PPFP outcomes.

| CONTEXT (C) | MECHANISM (M) | | OUTCOME (O) | CMOC |
|---|---|---|---|---|
| | **Resources** | **Reasoning** | | |
| *CONCEPT 2 –MOTIVATION* | | | | |
| **CMOC 6** Women with low motivation to improve health. | mHealth intervention content frame PPFP as beneficial to child health | Women are motivated by wanting to do the best for their child *(Facilitator)* | High engagement with mHealth programme | mHealth intervention content that frames PPFP as beneficial to child health (C) can lead to high engagement with the mHealth programme (O) because women want to do the best for their child (M). [48, 49, 51, 55, 57, 71, 72, 77] |
| **CMOC 7** Women enrolled in mHealth programme at periods of high intrinsic motivation (early pregnancy) | mHealth programme delivers gestational stage-based messaging | Women are motivated by the relevance of the messaging *(Facilitator)* | High engagement with mHealth programme | mHealth programmes that deliver gestational stage-based messaging early in pregnancy (C) can lead to high engagement with mhealth programmes (O) because women are motivated by the perceived relevance of the messaging (M). [50, 54–57, 60, 65, 72, 77] |
| **CMOC 8** Women want assistance and support during pregnancy from family and social circles | mHealth messages delivered in a shareable format (content or delivery mode) that helps family and social circles to assist or support women's engagement with the mHealth programme | Women highly value the messages *(Facilitator)* | High engagement with mHealth programme | Women who want assistance and social support during pregnancy get shareable mHealth messages (C) engagement with the mHealth programme is higher (O) because this helps the family and social circles to assist women's engagement with the programme (M). [16, 17, 50, 57, 58, 61, 66, 67, 70, 71, 76, 79] |
| **CMOC 9** Women receive family and social support to learn more about or take up PPFP | Women have a pre-existing interest in learning about PPFP methods/birth spacing | Women feel comfortable to engage with PPFP content in mHealth programme *(Facilitator)* | High engagement with mHealth programme | When women who have a pre-existing interest in learning about PPFP methods and birth spacing receive family and social support to learn about these topics (C), they will have high engagement with mHealth programmes (O) because women will feel more comfortable engaging with mHealth programmes that are accepted within their social support systems (M). [16, 17, 50, 57, 58, 61, 66, 67, 70, 71, 76, 79] |
| **CMOC 10** Women have pre-existing values, beliefs, or fears related to the use of modern contraceptive methods | mHealth messages promote modern contraceptive methods for PPFP | Women are unwilling to engage with the mHealth content that is not compatible with their values, beliefs, or fears *(Barrier)* | Little to no engagement with mHealth programme | Among women who have prior values, beliefs or fears related to the use of modern contraceptive methods (C), mHealth programme engagement is low (O) because mHealth messages promoting modern contraceptives are incompatible with these (M). [16, 17, 58, 60, 76, 79] |
| **CMOC 11** Women have not previously experienced adverse pregnancy and postpartum outcomes | mHealth messages promote PPFP methods to avoid inadequate birth spacing and associated adverse health outcomes. | Women do not feel these messages are applicable to them *(Barrier)* | Little to no engagement with mHealth programme | Women who have not previously experienced adverse health outcomes related to birth spacing or pregnancy in general (C) may not be motivated to engage with mHealth messages promoting PPFP methods (O) because they do not feel these messages are applicable to them (M). [49, 50, 52, 56, 62, 71, 72, 78, 79, 82] |

*(Continued)*

**Table 4.** (Continued)

| CONTEXT (C) | MECHANISM (M) | | OUTCOME (O) | CMOC |
|---|---|---|---|---|
| | Resources | Reasoning | | |
| **CMOC 12** Women with low intrinsic motivation to improve health | mHealth messages are tailored and personalized (e.g. to stage of pregnancy, culture, or context) | Women value the relevance of the messages *(Facilitator)* | High engagement with mHealth programme | When women with low intrinsic motivation about health issues receive mHealth messages that are tailored and personalized to their information needs (C) this can lead to high mHealth programme engagement (O) because women respond positively to timely and relevant health advice tailored to meet their cultural or pregnancy stage-based needs (M). [16, 15, 49, 51, 55, 60, 64, 72, 78, 79] |

with their husbands, who took the phone with them while they were out for work', thus impeding women's engagement with mHealth programmes [34]. Lefevre et al. suggests that phone sharing patterns within the Kilkari study population meant that men were often the primary listeners of the Kilkari calls and this could be beneficial in encouraging joint-decision making regarding PPFP [17].

Similarly, in the context of societies with traditional gender norms, mHealth programme content that is directed and targeting the needs of women may not yield changes in PPFP knowledge, awareness, or uptake [CMOC 5] [15, 52, 53, 57, 59, 63]. This is because men fear the negative consequences of their wives' independent mobile phone ownership and operation–such as infidelity–and gatekeep their wives' access to a shared device which leads to poor engagement with the mHealth programme and no change in PPFP knowledge, awareness, or uptake [15, 53, 57, 59, 63]. GSMA's report on Kilkari expands on the Scott et al. findings by reporting observations of husbands acting as gatekeepers of mobile phone access and many women had to obtain approval from husbands to subscribe to the Kilkari programme [52]. Women's sole ownership of a mobile phone is perceived as dangerous in certain cultural contexts due to the perceived susceptibility of women to sexual harassment from strangers through mobile phones [63]. Husbands often intervene and gatekeep access to mHealth interventions, as noted by Gupta et al. which can lead to reduced engagement by women in mHealth interventions and subsequently result in poor PPFP outcomes [53]. Scott et al.'s interview findings suggest that women with independent ownership of mobile phone are perceived negatively; assumptions are made regarding women's use of mobile phones for 'spurious purposes' and to 'indulge in obscene talks' [59].

**Facilitators of mHealth programme engagement.** Conversely, mHealth programme content directed and specifically targeting women was effective in motivating programme engagement and subsequent PPFP behaviour change in contexts where women receive family and social support for independent mobile phone ownership [CMOC 3] [52, 53, 55, 59, 74, 75, 80, 82]. Receiving family and social support for independent mobile phone ownership and operation makes women feel more confident to participate in mHealth programmes targeting her individual pregnancy and postpartum health needs, and can lead to greater mHealth programme engagement and subsequently improve PPFP uptake [52, 53, 55, 59, 74, 75, 80, 82]. Khatun suggests that mHealth programmes targeting the family unit will be more effective given that women are often 'still dependent on their husband's permission to seek healthcare through mobile phones' [82]. Encouraging social networks of support through increased interactions with health services, community health workers, and counsellors influence women's

continued engagement with mHealth programmes and can lead to higher PPFP use [53, 55, 74]. Increased interactions with support systems provide additional opportunities for health literacy improvement through dispelling 'myths and queries' related to PPFP, as well as providing reminders to better engage with health services promoting PPFP use.

## Motivation

**Facilitators of mHealth programme engagement.** The reviewed documents provided evidence of several pathways in which motivation is activated to produce high engagement with an mHealth programme. A woman's motivation to improve her pregnancy and postpartum health can be improved when mHealth intervention content frames PPFP as beneficial to child health and can subsequently lead to high engagement with the mHealth programme [CMOC 6] [48, 49, 51, 55, 57, 71, 72, 77]. Kocher et al. reported that women's primary motivation for behaviour change during pregnancy was 'the perceived benefit to the baby's health' and that the 'connection between their health and the health of the baby' was instrumental in activating the mechanism of motivation [71]. Women who felt the 'least personal responsibility for the health of their unborn child', were the least likely to change their behaviours related to pregnancy or postpartum care [72]. Jones et al. reported no significant difference in postpartum care seeking between the intervention and control groups pertaining to knowledge of newborn danger signs (OR 1.24, 95% CI 0.73–2.07) [55]. Similar levels of newborn care knowledge in intervention and control groups can be attributed to the overemphasis on 'observing the health of their children' among new mothers and healthcare providers, regardless of whether they receive intervention resources [55].

mHealth programmes that deliver gestational stage-based messaging are more likely to activate the mechanism of relevance when women are enrolled in the programme during periods of high intrinsic motivation, such as early into pregnancy [CMOC 7] [50, 54–57, 60, 65, 72, 77]. This leads to higher engagement with mHealth programmes and can potentially lead to improvements in PPFP knowledge, awareness, and uptake [50, 54–57, 60, 65, 72, 77]. Maslowsky et al. suggest that earlier intervention during pregnancy can help support women's transition to and continuation of health behaviours [56]. Lawrence and Haslam's work on smoking during pregnancy provides evidence for the context of intervention timing and its impact on activating the mechanism of motivation for behaviour change, which is usually highest during the first trimester [72]. Conversely, programmes that exclusively deliver PPFP messages during the postpartum period may benefit from two-way communication modes such as SMS messages between healthcare workers and participants [55, 66, 85].

mHealth messages delivered in a shareable format that helps family and friends to be involved can lead to higher engagement with mHealth programmes within certain contexts. In women who desire assistance and social support during the pregnancy period, such messages can lead to higher programme engagement because members of their social circle greatly value the content of these messages [CMOC 8] [16, 17, 50, 57, 58, 61, 66, 67, 70, 71, 76, 79]. SMS messages, voice-calls, and recordings are some examples of shareable mHealth delivery modes. Evidence from Dyer et al. and Bongaarts et al. suggest the importance of spousal acceptance of PPFP methods use on women's decision-making regarding the same concept [50]. Evidence from Jareethum et al. suggests that women who felt they were 'taken care of' experienced 'significantly increased confidence and decreased anxiety levels during the antenatal period' [67]. Receiving family and social support also interacted with women's pre-existing interest in learning about PPFP methods and influenced their comfort and confidence to engage in mHealth programmes which led to improved PPFP knowledge, awareness, and uptake [CMOC 9] [16, 17, 50, 57, 58, 61, 66, 67, 70, 71, 76, 79]. Murthy et al.'s findings report that

women 'shared and discussed the information they received' as a means of assessing 'buy-in' for behaviour change [58], further emphasizing the importance of family and social support.

Women with low intrinsic motivation to improve their health can become more motivated and engage effectively with mHealth programmes when content is tailored and personalised to their stage of pregnancy, or culture [CMOC 12] [16, 15, 49, 51, 55, 60, 64, 72, 78, 79, 86]. Women respond positively to tailored messages which improves their intrinsic motivation and can lead to higher engagement with the mHealth programme and subsequently improve PPFP knowledge, awareness, and uptake [15, 49, 51, 55, 16, 60, 64, 72, 78, 79, 86]. Examples of tailoring and personalization of mHealth messaging content include using multiple dialects, messages specific to rural communities, and messages framing medical advice through home remedies [51]. Increased contraceptive uptake in the Unger et al. RCT (1-way SMS: 72% and 2-way SMS: 73%; p = 0·03 and 0·02 versus 57% control, respectively) provides evidence of the effectiveness of personalized messaging on PPFP outcomes [60]. Scott et al. also implied that identifying and acknowledging the main barriers against PPFP use and incorporating these into tailored mHealth messages can strengthen the impact of mHealth programmes like Kilkari [16]. Tailoring mHealth programmes to individual women's needs through counselling and increased interactions with FLWs and health practitioners was also a key contextual influence on PPFP use [73].

**Barriers of mHealth programme engagement.** On the other hand, certain contexts interact with mHealth programmes and can deter women's engagement with mHealth programmes. mHealth messages that promote modern contraceptive methods will not effectively improve PPFP knowledge, awareness, or uptake when interacting with the context of women who have pre-existing values, beliefs, or fears related to the use of modern contraceptive methods [CMOC 10] [16, 17, 58, 60, 76, 79]. Women are unwilling to engage with content that is not compatible with their values, beliefs, or perceived needs, and can result in no change in PPFP uptake [16, 17, 58, 60, 76, 79]. Lefevre et al.'s Kilkari impact evaluation RCT found that men and women 'retained and appreciated' messages that aligned with their pre-existing beliefs and social norms related to PPFP but also 'overlooked or de-emphasised content that did not' [17]. Although FP was the most mentioned topic in the Kilkari calls, the messages were still too infrequent to effectively overturn entrenched norms regarding PPFP use; there was no significant difference in PPFP use between intervention and control arms of the Kilkari RCT (RR 1.03, 95% CI 0.97–1.09) [17]. Murthy et al.'s pseudo-randomized RCT reports no significant difference in PPFP use after delivery between the mMitra intervention group compared to controls (OR 0.78, 95% CI 0.58–1.07) [58]. Murthy et al. suggest that this statistically insignificant difference is attributable to high baseline awareness and knowledge of many mMitra topics including PPFP, but do not report ORs from sub-group analyses to this effect.

Experiences with adverse pregnancy and postpartum outcomes influenced women's motivation to engage with mHealth programmes. mHealth messages promoting PPFP methods as a means of avoiding short birth-spacing and the associated adverse health outcomes lead to limited programme engagement in contexts where women have not previously experienced negative pregnancy and postpartum outcomes because women do not perceive themselves as susceptible to adverse outcomes resulting from inadequate birth-spacing. Hence, they perceive the messages as not being applicable to them [CMOC 11] [49, 50, 52, 56, 62, 71, 72, 78, 79, 82]. Molla et al. reported that previous experiences were the greatest determinant of self-reported contraception usage, suggesting that unfamiliarity with contraception methods is a strong deterrent of engagement with PPFP messaging and subsequent PPFP uptake among women enrolled in mHealth interventions [76]. Kocher et al. reported that women did not 'perceive pregnancy as a weakened condition like illness' and believed it did not require unique considerations in terms of healthcare access or self-care measures [71]. Cleland et al. reported that

women perceived short birth-spacing as presenting only 'small risks' to health that they were 'prepared to take' given the prioritization of childbearing and the widespread availability of medical abortion methods if unintended pregnancies were to occur [62].

## Discussion

### Summary of findings

The empirical evidence reviewed in this study suggests that mHealth interventions are potentially more effective when barriers to phone access, use, and ownership, are absent and women's motivation for health-seeking behaviours is encouraged. mHealth message timing, tailored content, emphasis on child health, and perceptions of susceptibility to adverse pregnancy and postpartum outcomes were contexts in which the mechanisms were activated to increase engagement with mHealth programmes. Whether women could use, access, and/or own a mobile phone influenced which mechanism was activated or remained dormant. This realist review addresses the need for identifying and evaluating the components leading to intended and unintended consequences of mHealth interventions and aimed to explain how and why the interventions potentially lead to PPFP outcomes, thus building on the limitations outlined in both the Zulu et al. and Smith et al. systematic reviews [19, 20]. Since this review was intended to inform a realist evaluation of an existing mHealth programme, the evidence review was progressively focused to concentrate on the two key conceptual areas (namely 'Phone access, use, and ownership' and 'Motivation') that can be tested and further refined through the realist evaluation of mMitra. Tables 3 and 4, which are categorized into the domains of 'Phone access, use, and ownership' and 'Motivation', elaborate on the mechanisms underpinning the success and failure of mHealth interventions implemented in LMICs– specifically focussing on the important issue of engagement. The evidence in this review builds on Kabongo et al.'s realist review findings by suggesting that mechanisms which trigger mHealth programme engagement consequently led to the distal outcome of improvements in PPFP knowledge, awareness, or uptake [24]. However, the gap between mHealth programme engagement and PPFP knowledge, awareness, or uptake requires further research.

It is important to note that CMOs often overlap, and concepts can be defined as either contexts, mechanisms, or outcomes depending on the causal pathway under consideration. Women's low intrinsic motivation to improve their health is a context influencing their enrolment in mHealth programmes. The relevance of mHealth messages that improve women's motivation to learn about and change their PPFP behaviours is also a mechanism resulting in improved PPFP uptake and can be impacted by other contextual factors. Because of these overlapping definitions and the nature of complex social interventions, it is difficult to conclude from the evidence in this review that the causal pathways reflected by each CMOC occur in isolation. Although demographic context and mHealth programme features can act as an extrinsic motivator of PPFP behaviour change, women experience these contextual influences simultaneously. Thus, this review aimed to identify the full breadth of contextual influences on PPFP knowledge, awareness, and outcomes, and the mechanisms providing an explanation for how and why these influences work. Family support is one such complex phenomenon. From the findings of this review, familial involvement can activate women's feelings of support and empowerment to engage with mHealth programmes. Involvement by husbands and relatives can also serve to gatekeep women's engagement with mHealth programmes and potentially impede the distal outcome of PPFP uptake. Aung et al.'s systematic review presents contradicting evidence on the role of spousal involvement in improving PPFP uptake [18]; citing Harrington et al.'s sub-group analysis findings showing no difference in PPFP uptake based on spousal involvement [15]. Future research should identify both positive and negative

dimensions of familial involvement in mHealth interventions to determine their effects on PPFP knowledge acquisition and uptake.

The final programme theories build on the IPT by comparing technical features of mHealth programmes and their subsequent impact on PPFP outcomes. The review evidence strengthens the assumptions made in the IPT regarding the positive impact of timed messages aligning with women's stage of pregnancy and postpartum care while also proposing the importance of message formatting and how that allows women to assess social approval for PPFP behaviours. This finding aligns with the results of Abraham and Melendez-Torres' realist review of interventions targeting maternal health in LMICs by suggesting the importance of receiving family and social support to yield the outcome of health-service seeking and usage in pregnant women [87]. Additionally, the evidence in this review supports the importance of women's intrinsic motivation for behaviour change during pregnancy and the postpartum period and how mHealth interventions can capitalize and increase this motivation to improve PPFP outcomes through tailored messaging. A systematic review by Aung et al. on the effectiveness of mHealth interventions on family planning uptake in LMICs similarly found that motivational messaging was a key component driving family planning behaviour change [18]. It is important to note that women's intrinsic motivation for behaviour change largely applies to their interest in improving child health outcomes, which is a unique finding from this review. Although focusing women's health content through the lens of childcare and child-rearing can contradict the goals of female agency, and autonomy regarding healthcare decisions, this 'delicate' approach appears to improve PPFP use among women with lower motivation to improve their health and low engagement with health services. This finding aligns with published systematic review evidence by Aung et al. which suggested that tailored and motivational messaging were determinants of better PPFP outcomes [18]. The included studies suggest that there are differing degrees of digital literacy; phone calls require less familiarity with mobile devices compared to SMS messages and may cater to the needs of different participant demographics. The Aponjon programme was initially delivered using SMS messages but formative field studies suggested there was an 'unmet demand for voice calls in rural areas and among people of the lowest socioeconomic status' [34]. Additionally, as the review evidence suggests, gender norms influence the level of support women receive for phone ownership and utilisation which affects their abilities to adequately receive mHealth intervention resources and can result in low engagement as well as low knowledge, awareness, and uptake of PPFP. This finding builds on the Zulu et al. systematic review which identified the importance of tailoring intervention messaging to account for gender and social norms in order to be contextually relevant and effective at improving PPFP uptake [20].

## Strengths and limitations

The primary strength of this review is the use of rigorous step-by-step procedures for screening and theory refinement. All titles and abstracts were double screened and emerging programme theories were discussed with experts in the fields of mHealth, family planning, and realist methodologies. Discussions with experts and incorporation of their feedback into the review process helped with sense-checking and adequately abstracting the theories to provide a middle-range explanation that is likely to be applied across contexts of mHealth programme implementation. Discussions between co-authors during the double screening process were helpful in refining and focusing the scope of the review through consideration of varying components of mHealth programme delivery. A strength of this study is the use of this review's findings to inform a realist evaluation of the mMitra programme. Narrowing the scope of explored programme theories to focus on hypotheses that can be tested (confirmed, refuted or

refined) using real-world evidence from an established mHealth programme in an LMIC setting is a unique contribution of this study and serves to strengthen the findings proposed in this review. Another strength of this review is the inclusion of both peer-reviewed and grey literature to build programme theories. For complex social interventions such as mHealth programmes, programme reports, and grey literature provide knowledge and expertise on programme implementation that is often absent from peer-reviewed literature. While peer-reviewed literature were more likely to report outcomes, programme documents were useful for identifying contexts outside of participant demographics such as mobile network infrastructure, programme funding, as well as language and mode of mHealth programme delivery.

One limitation of this review is the certainty of the conclusions drawn from the included evidence, given that few included studies lacked methodological rigour, as assessed by the MMAT tool. Realist reviews do not intend to discriminate against the rigour of evidence included in the synthesis and prioritise the relevance of evidence pertaining to contexts, mechanisms, and outcomes that a study can provide. Although the included studies contributed in-depth information on contexts and mechanisms, the low methodological rigour of some included quantitative studies made us question the strength of evidence supporting improved PPFP outcomes in their study populations. Unblinded randomization processes and poor adherence to the assigned intervention in the Murthy et al. study, for instance, makes it difficult to conclude with certainty whether or not the mMitra programme was effective in improving the adoption of family planning methods in the study population. Nonetheless, the included studies were appropriate for identifying a wide range of possible CMOs which fulfilled the aim of this review. Another potential limitation of this study is the lack of diversity in LMIC settings represented by the included studies. This review includes evidence primarily from India and Kenya, where a large proportion of mHealth interventions within the realm of RMNCH are implemented. Although this review aimed to iteratively use 'all relevant available evidence' [31] from peer-reviewed literature and programme documents for CMOC development, the full breadth of evidence from reproductive health organisations was not explored in this study. The lack of guidance on methodically reviewing grey literature contributes to this limitation. Future studies could improve rigour by increasing transparency of reporting and taking greater steps to protect against bias in exposure and outcome measurement, randomisation, and adherence to the interventions.

Most mHealth interventions evaluated in this review had some commercial funding (Table 2). Some of these funders include Dnet, Johnson & Johnson, and the Vodacom Foundation. Although many included documents explicitly state commercial funding sources such as the aforementioned pharmaceutical companies and mobile network operators, none of the included documents detail the impact of these funding structures on programme implementation and impact [88, 89]. The commercial aspects of mobile network connectivity in many LMIC settings is inextricably linked with access to mobile phones and should be further explored as a context in future reviews of mHealth evidence.

## Conclusions

To our knowledge, this is the first realist review exploring the impact of mHealth interventions implemented in LMICs on postpartum family planning outcomes. This study identified CMOs representing a diverse range of concepts crucial to the implementation and effectiveness of mHealth programmes such as programme design and delivery, socio-demographics, complex social norms, and phone ownership practices. The unique contribution of this review is the evidence provided for these contexts and their underlying mechanisms as well as how these pathways hypothetically function in real-world settings. The CMOCs are applicable

across a range of settings. Our findings suggest the importance of activating women's intrinsic motivation for health behaviour change during early pregnancy to improve PPFP outcomes. Also, mHealth intervention delivery should address different phone ownership, take account of societal norms and sharing patterns to improve programme engagement. Our CMOCs provide a foundation for developing hypotheses on the effectiveness of mHealth implementation contexts on PPFP outcomes. The accompanying mixed-methods realist evaluation of mMitra will further expand on this review's CMOC concepts and provide insight into the integration of realist reviews and realist evaluations in implementation research.

## Supporting information

**S1 Text. Detailed information on the mMitra programme.**
(DOCX)

**S1 Fig. Dalkin et al. framework diagram.**
(DOCX)

**S1 Table. Realist review systematic search strategy.**
(DOCX)

**S1 Data. Quality appraisal of included documents using MMAT tool and mERA appraisal tool.**
(DOCX)

**S1 File. Realist review reporting standards from RAMESES project.**
(PDF)

**S1 Checklist. Completed PRISMA checklist.**
(DOCX)

## Author Contributions

**Conceptualization:** Abinaya Chandrasekar, Geoffrey Wong, Ona McCarthy.

**Data curation:** Abinaya Chandrasekar, Judie Mbogua, Esther Curtin, Ursula Gazeley.

**Formal analysis:** Abinaya Chandrasekar.

**Investigation:** Abinaya Chandrasekar.

**Methodology:** Abinaya Chandrasekar, Emily Warren, Caroline Free, Kathryn Church, Ona McCarthy.

**Project administration:** Abinaya Chandrasekar.

**Supervision:** Emily Warren, Caroline Free, Ona McCarthy.

**Validation:** Emily Warren, Caroline Free, Geoffrey Wong, Kathryn Church, Ona McCarthy.

**Visualization:** Ona McCarthy.

**Writing – original draft:** Abinaya Chandrasekar.

**Writing – review & editing:** Abinaya Chandrasekar, Emily Warren, Caroline Free, Judie Mbogua, Esther Curtin, Ursula Gazeley, Geoffrey Wong, Kathryn Church, Ona McCarthy.

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
