## [Decision Letter · Decision Letter 0]

14 May 2024

PGPH-D-23-02613

mHealth interventions for postpartum family planning in LMICs: A Realist Review

Dear Dr. Chandrasekar,

Thank you for submitting your manuscript to PLOS Global Public Health. After careful consideration, we feel that it has merit but does not fully meet PLOS Global Public Health’s publication criteria as it currently stands. Therefore, we invite you to submit a revised version of the manuscript that addresses the points raised during the review process.

At minimum, please **address** the following comments for acceptance:

1.There is scope to make the introduction crisper by tightening it.

2.Please add sub-headings to clearly state challengers/barriers and facilitators to uptake of the technology.

3.Clarify inclusion criteria in the methods section.

4.Highlight if engagement in mHealth leads to subsequent behavior change regarding uptake of PPFP. It would be helpful to mention this briefly in the abstract and make it a greater point of emphasis throughout the manuscript as ultimately, we want mHealth to drive health outcomes.

5.Since PGH has a wider audience, please define unmet need for modern contraception

6.Please add appropriate citations and justify why addressing the social and behavioral barriers to access should prove effective to increase uptake of PPFP.

7.Please clarify what exactly is the mixed evidence of mHealth intervention effectiveness in the RMNCH landscape? Please cite specific studies that showcase ineffective interventions and the respective outcomes they examine.

8.Add details on how differential weight was given to better designed studies.

9.In the PICOS table, clearly define indicators for “PPFP outcomes”– what counts as a PPFP outcome and what does not? What PPFP outcomes can mHealth interventions realistically improve?

10.In the discussion, please outline what made some studies low in methodological rigor and suggestions for what future studies can incorporate to avoid making the same mistakes. This will also help the readers understand how strongly to consider such evidence.

11.Correct all typographical errors and spell check your manuscript.

12. Situate your findings in the discussion section by comparing your findings to other similar reviews. 

13. Initial Programme Theory – please add a clarification on how you decided on this particular program and these documents for extracting the context–mechanism–outcome configurations? Add some justification for what makes them so well suited for this task. I agree with reviewer 2 that this also seems like this program is the core focus of the review, which should be discussed earlier in the introduction given how central it is to the manuscript. This isn’t currently clear in the introduction or abstract.

We look forward to receiving your revised manuscript.

Kind regards,

Lakshmi Gopalakrishnan, PhD

Academic Editor

Journal Requirements:

Additional Editor Comments (if provided): None

Reviewers' comments:

Reviewer's Responses to Questions

**Comments to the Author**

1. Does this manuscript meet PLOS Global Public Health’s publication criteria? Is the manuscript technically sound, and do the data support the conclusions? The manuscript must describe methodologically and ethically rigorous research with conclusions that are appropriately drawn based on the data presented.

Reviewer #1: Yes

Reviewer #2: Yes

2. Has the statistical analysis been performed appropriately and rigorously?

Reviewer #1: Yes

Reviewer #2: Yes

3. Have the authors made all data underlying the findings in their manuscript fully available (please refer to the Data Availability Statement at the start of the manuscript PDF file)?

Reviewer #1: Yes

Reviewer #2: Yes

4. Is the manuscript presented in an intelligible fashion and written in standard English?

Reviewer #1: Yes

Reviewer #2: Yes

5. Review Comments to the Author

Reviewer #1: This review provides a comprehensive examination of mHealth interventions in Low and Middle-Income Countries , with a focus on their impact on postpartum family planning (PPFP) outcomes. The study presents clear findings and draws connections between different components of mHealth programs and their effects on PPFP knowledge, awareness, and uptake.

- The introduction needs to be presented in a succint manner. Currently it reads like the introduction of a thesis.

- The findings need robust sub-headings to clearly state challengers/barriers and facilitators to uptake of the technology.

- The results need to be re-written to not include points that are suitable for the discussion section.

- A more critical discussion on the implications of this limitation for the overall findings would add depth to the review.

- The review does not delve into the impact of funding structures on program implementation. Further exploration of the commercial aspects of mobile network connectivity and its implications for mHealth interventions in LMICs would add depth to the analysis.

- Lack of comparison to similar reviews in the discussion section. Authors would beenfit from restructuring the paper to enhance the discussion, formulate a succinct introduction and structure the results better for improved readability.

Reviewer #2: This is a well-written manuscript with proper analyses conducted. The authors make use of the published and grey literature, so all of the data is provided from these other manuscripts. The data also support the conclusions. My full comments are available in the attached document.

6. PLOS authors have the option to publish the peer review history of their article (what does this mean?). If published, this will include your full peer review and any attached files.

**Do you want your identity to be public for this peer review?** For information about this choice, including consent withdrawal, please see our Privacy Policy.

Reviewer #1: No

Reviewer #2: No

---

## [Editor Report · Decision Letter 1]

28 Jun 2024

mHealth interventions for postpartum family planning in LMICs: A Realist Review

PGPH-D-23-02613R1

Dear Miss Chandrasekar,

We are pleased to inform you that your manuscript 'mHealth interventions for postpartum family planning in LMICs: A Realist Review' has been provisionally accepted for publication in PLOS Global Public Health.

Best regards,

Lakshmi Gopalakrishnan, PhD

Academic Editor